# Improved Word Representation Learning with Sememes

## Abstract

Sememes are minimum semantic units of word meanings, and the meaning of each word sense is typically composed by several sememes. Since sememes are not explicit for each word, people manually annotate word sememes and form linguistic common-sense knowledge bases. In this paper, we present that, word sememe information can improve word representation learning (WRL), which maps words into a low-dimensional semantic space and serves as a fundamental step for many NLP tasks. The key idea is to utilize word sememes to accurately capture exact meanings of a word within specific contexts. More specifically, we follow the framework of Skip-gram and present three sememe-encoded models to learn representations of sememes, senses and words, where we apply the attention scheme to detect word senses in various contexts. We conduct experiments on two tasks including word similarity and word analogy, and our models significantly outperform baselines. The results indicate that WRL can benefit from sememes via the attention scheme, and also confirm our models being capable of properly modeling sememe information.

## 1 Introduction

Sememes are defined as minimum semantic units of word meanings, and there exists a limited close set of sememes to compose the semantic meanings of an open set of concepts (i.e. word sense). However, sememes are not explicit for each word. Hence, people manually annotate word sememes and build linguistic common-sense knowledge bases.

HowNet (Dong and Dong, 2003) is one of such knowledge bases, which annotates each concepts in Chinese with one or more relevant sememes. Different from WordNet (Miller, 1995), the philosophy of HowNet emphasizes the significance of `part` and `attribute` represented by sememes. HowNet has been widely utilized in word similarity computation (Liu and Li, 2002) and sentiment analysis (Xianghua et al., 2013), and in section 3.2 we will give detailed introduction on sememes, senses and words in HowNet.

In this paper, we aim to incorporate word sememes into word representation learning (WRL) and learn improved word embeddings in a low-dimensional semantic space. WRL is a fundamental and critical step in many NLP tasks such as language modeling (Bengio et al., 2003) and neural machine translation (Sutskever et al., 2014).

There have been a lot of researches for learning word representations, among which word2vec (Mikolov et al., 2013) achieves a nice balance between effectiveness and efficiency. In word2vec, each word corresponds to one single embedding, ignoring the polysemy of most words. To address this issue, (Huang et al., 2012) introduces multi-prototype model for WRL, conducting unsupervised word sense induction and embeddings according to context clusters. (Chen et al., 2014) further utilizes the `synset` information in WordNet to instruct word sense representation learning.

From these previous studies we conclude that, word sense disambiguation are critical for WRL, and we believe that the sememe annotation of word senses in HowNet can provide essential semantic regularization for the both tasks. To explore its feasibility, we propose a novel **S**ememe-**E**ncoded **W**ord **R**epresentation **L**earning (SE-WRL) model, which detects word senses and learns representations simultaneously. More

specifically, this framework regards each word sense as a combination of its sememes, and iteratively performs word sense disambiguation according to their contexts and learn representations of sememes, senses and words by extending Skip-gram in word2vec (Mikolov et al., 2013). In this framework, an attention-based method is proposed to automatically select appropriate word senses according to contexts. To take full advantages of sememes, we propose three different learning and attention strategies for SE-WRL.

In experiments, we evaluate our framework on two tasks including word similarity and word analogy, and further conduct case studies on sememe, sense and word representations. The evaluation results show that our models outperform other baselines significantly, especially on word analogy. This indicates that our models can automatically detect appropriate word senses according to contexts, and both word sense disambiguation and representation learning can benefit from the sememe annotation in HowNet.

The key contributions of this work are concluded as follows: (1) To the best of our knowledge, this is the first work to utilize sememes in HowNet to improve word representation learning. (2) We successfully apply the attention scheme to detect word senses and learn representations according to contexts with the favor of the sememe annotation in HowNet. (3) We conduct extensive experiments and verify the effectiveness of incorporating word sememes for improved WRL.

## 2 Related Work

### 2.1 Word Representation

Recent years have witnessed the great thrive in word representation learning. It is simple and straightforward to represent words using one-hot representations, but it usually struggles with the data sparsity issue and the neglect of semantic relations between words.

To address these issues, (Rumelhart et al., 1988) proposes the idea of distributed representation which projects all words into a continuous low-dimensional semantic space, considering each word as a vector. Distributed word representations are powerful and have been widely utilized in many NLP tasks, including neural language models (Bengio et al., 2003; Mikolov et al., 2010), machine translation (Sutskever et al., 2014; Bahdanau et al., 2015), parsing (Chen and Manning, 2014)

and text classification (Zhang et al., 2015). Word distributed representations are capable of encoding semantic meanings in vector space, serving as the fundamental and essential inputs of many NLP tasks.

There are large amounts of efforts devoted to learning better word representations. As the exponential growth of text corpora, model efficiency becomes an important issue. (Mikolov et al., 2013) proposes two models, CBOW and Skip-gram, achieving a good balance between effectiveness and efficiency. These models assume that the meanings of words can be well reflected by their contexts, and learn word representations by maximizing the predictive probabilities between words and their contexts. (Pennington et al., 2014) further utilizes matrix factorization on word affinity matrix to learn word representations. However, these models merely arrange only one vector for each word, regardless of the fact that many words have multiple senses. (Huang et al., 2012; Tian et al., 2014) utilize multi-prototype vector models to learn word representations, and build distinct vectors for each word sense. (Neelakantan et al., 2015) presents an extension to Skip-gram model for learning non-parametric multiple embeddings per word. (Rothe and Schütze, 2015) also utilizes an Autoencoder to jointly learn word, sense and synset representations in the same semantic space.

This paper, for the first time, jointly learns representations of sememes, senses and words. The sememe annotation in HowNet provides useful semantic regularization for WRL. Moreover, the unified representations incorporated with sememes also provide us more explicit explanations of both word and sense embeddings.

### 2.2 Word Sense Disambiguation and Representation Learning

Word sense disambiguation (WSD) aims to computationally identify word senses or meanings in a certain context. There are mainly two approaches for WSD, namely the supervised methods and the knowledge-based methods. Supervised methods usually take the surrounding words or senses as features and use classifiers like SVM for word sense disambiguation (Lee et al., 2004), which are intensively limited to the time-consuming human annotation of training data.

On contrary, knowledge-based methods utilize large external knowledge resources such as knowl-

edge bases or dictionaries to suggest possible senses for a word. (Banerjee and Pedersen, 2002) exploits the rich hierarchy of semantic relations in WordNet (Miller, 1995) for an adapted dictionary-based WSD algorithm. (Bordes et al., 2011) introduces `synset` information in WordNet to WRL. (Chen et al., 2014) considers synsets in WordNet as different word senses, and jointly conducts word sense disambiguation and word / sense representation learning. (Guo et al., 2014) considers bilingual datasets to learn sense-specific word representations. Moreover, (Jauhar et al., 2015) proposes two approaches to learn sense-specific word representations that are grounded to ontologies. (Pilehvar and Collier, 2016) utilizes personalized PageRank to learn de-conflated semantic representations of words.

In this paper, we follow the knowledge-based approach and automatically detect word senses according to the contexts with the favor of sememe information in HowNet. To the best of our knowledge, this is the first attempt to apply attention-based models to encode sememe information for word representation learning.

## 3 Methodology

In this section we present our framework Sememe-Encoded WRL (SE-WRL) that considers sememe information for word sense disambiguation and representation learning. Specifically, we learn our models on a large-scale text corpus with the semantic regularization of the sememe annotation in HowNet, and obtain sememe, sense and word embeddings for evaluation tasks.

In the following sections, we first introduce HowNet and the structures of sememes, senses and words. Then we discuss the conventional WRL model Skip-gram that we utilize for the the sememe-encoded framework. Finally, we propose three sememe-encoded models in details.

### 3.1 Sememes, Senses and Words in HowNet

In this section, we first introduce the arrangement of sememes, senses and words in HowNet. HowNet annotates precise senses to each word, and for each sense HowNet annotates the significance of parts and attributes represented by sememes.

Fig. 1 gives an example of sememes, senses and words in HowNet. The first layer represents the **word** "apple". The word "apple" actually has two main **senses** shown on the second layer: one is a sort of juicy fruit (*apple*), and another is a famous computer brand (*Apple brand*). The third and following layers are those **sememes** explaining each sense. For instance, the first sense *Apple brand* indicates a computer brand, and thus has sememes `computer`, `bring` and `SpeBrand`.

From Fig. 1 we can find that, sememes of many senses in HowNet are annotated with various relations, such as *define* and *modifier*, and form complicated hierarchical structures. In this paper, for simplicity we only consider all annotated sememes of each sense as a sememe set without considering their internal structure. HowNet assumes the limited annotated sememes can well represent senses and words in real-world scenario, and thus sememes are expected to be useful for both WSD and WRL.

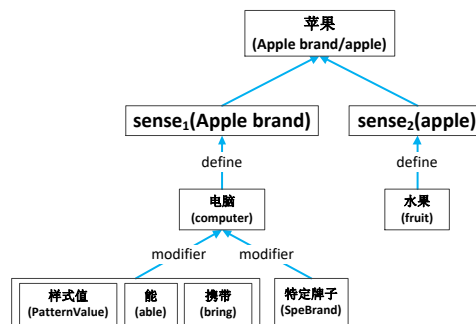

Figure 1: Examples of sememes, senses and words.

We introduce the notions utilized in the following sections as follows. We define the overall sememe, sense and word sets used in training as $X$, $S$ and $W$ respectively. For each $w \in W$, there are possible multiple senses $s_i^{(w)} \in S^{(w)}$ where $S^{(w)}$ represents the sense set of $w$. Each sense $s_i^{(w)}$ consists of several sememes $x_j^{(s_i)} \in X_i^{(w)}$. For each target word $w$ in a sequential plain text, $C(w)$ represents its context word set.

### 3.2 Conventional Skip-gram Model

We directly utilize the widely-used model Skip-gram to implement our SE-WRL model, because Skip-gram has well balanced effectiveness as well as efficiency (Mikolov et al., 2013). The standard skip-gram model assumes that word embeddings should relate to their context words. It aims at maximizing the predictive probability of context words conditioned on the target word $w$. For-

mally, we utilize a sliding window to select the context word set $C(w)$. For a word sequence $H = \{w_1, \cdots, w_n\}$, Skip-gram model intends to maximize:

$$L(H) = \sum_{i=K}^{n-K} \log \Pr(w_{i-K}, \cdots, w_{i+K}|w_i), \quad (1)$$

where $K$ is the size of sliding window. $\Pr(w_{i-K}, \cdots, w_{i+K}|w_i)$ represents the predictive probability of context words conditioned on the target word $w_i$, formalized by the following softmax function:

$$\Pr(w_{i-K}, \cdots, w_{i+K}|w_i) = \prod_{w_c \in C(w_i)} \Pr(w_c|w_i)$$

$$= \prod_{w_c \in C(w_i)} \frac{\exp(\mathbf{w}_c^\top \cdot \mathbf{w}_i)}{\sum_{w_i' \in W} \exp(\mathbf{w}_c^\top \cdot \mathbf{w}_i')}, \quad (2)$$

in which $\mathbf{w}_c$ and $\mathbf{w}_i$ stand for embeddings of context word $w_c \in C(w_i)$ and target word $w_i$ respectively. We can also follow the strategies of hierarchical softmax and negative sampling proposed in (Mikolov et al., 2013) to accelerate the calculation of softmax.

### 3.3 SE-WRL Model

In this section, we introduce the SE-WRL models with three different strategies to utilize sememe information, including Simple Sememe Aggregation Model (SSA), Sememe Attention over Context Model (SAC) and Sememe Attention over Target Model (SAT).

#### 3.3.1 Simple Sememe Aggregation Model

The Simple Sememe Aggregation Model (SSA) is a straightforward idea based on Skip-gram model. For each word, SSA considers all sememes in all senses of the word together, and represents the target word using the average of all its sememe embeddings. Formally, we have:

$$\mathbf{w} = \frac{1}{m} \sum_{s_i^{(w)} \in S^{(w)}} \sum_{x_j^{(s_i)} \in X_i^{(w)}} \mathbf{x}_j^{(s_i)}, \quad (3)$$

which means the word embedding of $w$ is composed by the average of all its sememe embeddings. Here, $m$ stands for the overall number of sememes belonging to $w$.

This model simply follows the assumption that, the semantic meaning of a word is composed by

the semantic units, i.e., sememes. As compared to the conventional Skip-gram model, since sememes are shared by multiple words, this model can utilize sememe information to encode latent semantic correlations between words. In this case, similar words that share the same sememes may finally obtain similar representations.

#### 3.3.2 Sememe Attention over Context Model

The SSA Model replaces the target word embedding with the aggregated sememe embeddings to encode sememe information into word representation learning. However, each word in SSA model still has only one single representation in different contexts, which cannot deal with polysemy of most words. It is intuitive that we should construct distinct embeddings for a target word according to specific contexts, with the favor of word sense annotation in HowNet.

To address this issue, we come up with the Sememe Attention over Context Model (SAC). SAC utilizes the attention scheme to automatically select appropriate senses for context words according to the target word. That is, SAC conducts word sense disambiguation for context words to learn better representations of target words. The structure of the SAC model is shown in Fig. 2.

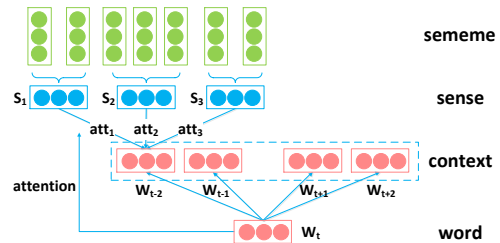

Figure 2: Sememe Attention over Context Model.

More specifically, we utilize the original word embedding for target word $w$, but use sememe embeddings to represent context word $w_c$ instead of original context word embeddings. Suppose a word typically demonstrates some specific senses in one sentence. Here we employ the target word embedding as an attention to select the most appropriate senses to make up context word embeddings. We formalize the context word embedding $\mathbf{w}_c$ as follows:

$$\mathbf{w}_c = \sum_{j=1}^{|S^{(w_c)}|} att(s_j^{(w_c)}) \cdot \mathbf{s}_j^{(w_c)}, \quad (4)$$

where $\mathbf{s}_j^{(w_c)}$ stands for the $j$-th sense embedding of $w_c$, and $att(s_j^{(w_c)})$ represents the attention score of the $j$-th sense with respect to the target word $w$, defined as follows:

$$att(s_j^{(w_c)}) = \frac{\exp(\mathbf{w} \cdot \hat{\mathbf{s}}_j^{(w_c)})}{\sum_{k=1}^{|S^{(w_c)}|} \exp(\mathbf{w} \cdot \hat{\mathbf{s}}_k^{(w_c)})}. \qquad (5)$$

Note that, when calculating attention, we use the average of sememe embeddings to represent each sense $s_j^{(w_c)}$:

$$\hat{\mathbf{s}}_j^{(w_c)} = \frac{1}{|X_j^{(w_c)}|} \sum_{k=1}^{|X_j^{(w_c)}|} \mathbf{x}_k^{(s_j)}. \qquad (6)$$

The attention strategy assumes that, the more relevant a context word sense embedding is to the target word $\mathbf{w}$, the more this sense should be considered when building context word embeddings. With the favor of attention scheme, we can represent each context word as a certain distribution over its sense. This can be regarded as soft WSD. As shown in experiments, it will help learn better word representations.

### 3.3.3 Sememe Attention over Target Model

The Sememe Attention over Context Model can flexibly select appropriate senses and sememes for context words according to the target word. The process can be also applied to select appropriate senses for the target word, by taking context words as attention. Hence, we propose the Sememe Attention over Target Model (SAT) as shown in Fig. 3.

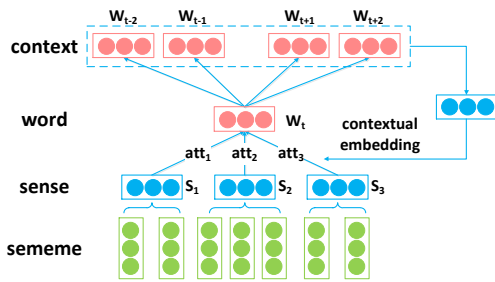

Figure 3: Sememe Attention over Target Model.

Different from SAC model, SAT learns the original word embeddings for context words, but sememe embeddings for target words. We apply context words as attention over multiple senses of the target word $w$ to build the embedding of $w$, formalized as follows:

$$\mathbf{w} = \sum_{j=1}^{|S^{(w)}|} att(s_j^{(w)}) \cdot \mathbf{s}_j^{(w)}, \qquad (7)$$

where $\mathbf{s}_j^{(w)}$ stands for the $j$-th sense embedding of $w$, and the context-based attention is defined as follows:

$$att(s_j^{(w)}) = \frac{\exp(\mathbf{w}_c' \cdot \hat{\mathbf{s}}_j^{(w)})}{\sum_{k=1}^{|S^{(w)}|} \exp(\mathbf{w}_c' \cdot \hat{\mathbf{s}}_k^{(w)})}, \qquad (8)$$

where, similar to Eq. (6), we also use the average of sememe embeddings to represent each sense $s_j^{(w)}$. Here, $\mathbf{w}_c'$ is the context embedding, consisting of a constrained window of word embeddings in $C(w_i)$. We have:

$$\mathbf{w}_c' = \frac{1}{2K'} \sum_{k=i-K'}^{k=i+K'} \mathbf{w}_k, \quad k \neq i. \qquad (9)$$

Note that, since in experiment we find the sense selection of the target word only relies on more limited context words for calculating attention, hence we select a smaller $K'$ as compared to $K$.

Recall that, SAC only uses one target word as attention to select senses of context words, but SAT use several context words together as attention to select appropriate senses of target words. Hence SAT is expected to conduct more reliable WSD and result in more accurate word representations, which will be explored in experiments.

## 4 Experiments

In this section, we evaluate the effectiveness of our SE-WRL models on two tasks including word similarity and word analogy, which are two classical evaluation tasks mainly focusing on evaluating the quality of learned word representations. We also explore the potential of our models in word sense disambiguation with case study, showing the power of our attention-based models.

### 4.1 Dataset

We use the webpages in Sogou-T[1] as the text corpus to learn WRL models. Sogou-T is provided by a Chinese commercial search engine, which contains 2.7 billion words in total.

---
[1] https://www.sogou.com/labs/resource/t.php

We also utilize the sememe annotation in HowNet. The number of distinct sememes used in this paper is $1,889$. The average senses for each word is about $2.4$, while the average sememes for each sense is about $1.6$. Throughout the Sogou-T corpus, we find that $42.2\%$ of words have multiple senses. This indicates the significance of WSD.

For evaluation, we choose wordsim-240 and wordsim-297[2] to evaluate the performance of word similarity computation. The two datasets both contain frequently-used Chinese word pairs with similarity scores annotated manually. We choose the Chinese Word Analogy dataset proposed by (Chen et al., 2015) to evaluate the performance of word analogy inference, that is, $\mathbf{w}(\text{``king''}) - \mathbf{w}(\text{``man''}) \simeq \mathbf{w}(\text{``queen''}) - \mathbf{w}(\text{``woman''})$.

## 4.2 Experimental Settings

We evaluate three SE-WRL models including S-SA, SAC and SAT on all tasks. As for baselines, we consider three conventional WRL models including Skip-gram, CBOW and GloVe. For Skip-gram and CBOW, we directly use the code released by Google (Mikolov et al., 2013). GloVe is proposed by (Pennington et al., 2014), which seeks the advantages of the WRL models based on statistics and those based on prediction. Moreover, we implement another baseline Maximum Selection over Target Model (MST) inspired by (Chen et al., 2014). It represents the current word embeddings with only the most probable sense according to the contexts, instead of viewing a word as a certain distribution over all its senses similar to that of SAT.

For a fair comparison, we train these models with the same experimental settings and with their best parameters. As for the parameter settings, we set the context window size $K = 8$ as the upper bound, and during training the window size is dynamically selected ranging from $1$ to $8$ randomly. We set the dimensions of word, sense and sememe embeddings to be the same $200$. For learning rate $\alpha$, its initial value is $0.025$ and will descend through iterations. We set the number of negative samples to be $25$. We also set a lower bound of word frequency as $50$, and in the training set those words less frequent than this bound will be filtered out. For SAT, we set $K' = 2$.

[2]https://github.com/Leonard-Xu/CWE/tree/master/data

## 4.3 Word Similarity

The task of word similarity aims to evaluate the quality of word representations by comparing the similarity ranks of word pairs computed by WRL models with the ranks given by dataset. WRL models typically compute word similarities according to their distances in the semantic space.

### 4.3.1 Evaluation Protocol

In experiments, we choose the cosine similarity between two word embeddings to rank word pairs. For evaluation, we compute the Spearman correlation between the ranks of models and the ranks of human judgements.

| Model | Wordsim-240 | Wordsim-297 |
|---|---|---|
| CBOW | 57.7 | 61.1 |
| GloVe | 59.8 | 58.7 |
| Skip-gram | 58.5 | 63.3 |
| MST | 59.2 | 62.8 |
| SSA | 58.9 | 64.0 |
| SAC | 59.0 | 63.1 |
| SAT | **63.2** | **65.6** |

Table 1: Evaluation results of word similarity computation.

### 4.3.2 Experiment Results

Table 1 shows the results of these models for word similarity computation. From the results we can observe that:

(1) Our SAT model outperforms other models, including all baselines, on both two test sets. This indicates that, by utilizing sememe annotation properly, our model can better capture the semantic relations of words, and learn more accurate word embeddings.

(2) The SSA model represents a word with the average of its sememe embeddings. In general, S-SA model performs slightly better than baselines, which tentatively proves that sememe information is helpful. The reason is that words which share common sememe embeddings will benefit from each other. Especially, those words with lower frequency, which cannot be learned sufficiently using conventional WRL models, in contrast can obtain better word embeddings from SSA simply because their sememe embeddings can be trained sufficiently through other words.

(3) The SAT model performs much better than SSA and SAC. This indicates that SAT can obtain more precise sense distribution of a word. The reason has been mentioned above that, different from

| Model | Accuracy | | | | Mean Rank | | | |
|---|---|---|---|---|---|---|---|---|
| | Capital | City | Relationship | All | Capital | City | Relationship | All |
| CBOW | 49.8 | 85.7 | **86.0** | 64.2 | 36.98 | 1.23 | 62.64 | 37.62 |
| GloVe | 57.3 | 74.3 | 81.6 | 65.8 | 19.09 | 1.71 | 3.58 | 12.63 |
| Skip-gram | 66.8 | 93.7 | 76.8 | 73.4 | 137.19 | 1.07 | 2.95 | 83.51 |
| MST | 65.7 | 95.4 | 82.7 | 74.5 | 50.29 | 1.05 | 2.48 | 31.05 |
| SSA | 62.3 | 93.7 | 81.6 | 71.9 | 45.74 | 1.06 | 3.33 | 28.52 |
| SAC | 61.6 | 95.4 | 77.9 | 70.8 | 19.08 | 1.02 | **2.18** | 12.18 |
| SAT | **83.2** | **98.9** | 82.4 | **85.3** | **14.42** | **1.01** | 2.63 | **9.48** |

Table 2: Evaluation results of word analogy inference.

SAC using only one target word as attention for WSD, SAT adopts richer contextual information as attention for WSD.

(4) SAT works better than MST, and we can conclude that, a soft disambiguation over senses prevents inevitable errors when selecting only one most-probable sense. The result makes sense because for many words, their various senses are not always completely different from each other, but share some common elements. In some contexts, a single sense may not convey the exact meaning of this word.

## 4.4 Word Analogy

Word analogy inference is another widely-used task to evaluate the quality of WRL models (Mikolov et al., 2013).

### 4.4.1 Evaluation Protocol

The dataset proposed by (Chen et al., 2015) consists of $1,124$ analogies, which contains three analogy types: (1) capitals of countries (Capital), 677 groups; (2) states/provinces of cities (City), 175 groups; (3) family words (Relationship), 272 groups. Given an analogy group of words $(w_1, w_2, w_3, w_4)$, WRL models usually get $\mathbf{w}_2 - \mathbf{w}_1 + \mathbf{w}_3$ equal to $\mathbf{w}_4$. Hence for word analogy inference, we suppose $w_4$ is missing, and WRL models will rank all candidate words according to their scores as follows:

$$R(w) = \cos(\mathbf{w}_2 - \mathbf{w}_1 + \mathbf{w}_3, \mathbf{w}), \qquad (10)$$

and select the top-ranked word as the answer.

For word analogy inference, we consider two evaluation metrics: (1) **Accuracy**. For each analogy group, a WRL model selects the top-ranked word $w = \arg\max_w R(w)$, which is judged as positive if $w = w_4$. The percentage of positive samples is regarded as the accuracy score for this WRL model. (2) **Mean Rank**. For each analogy group, a WRL model will assign a rank for

the gold standard word $w_4$ according to the scores computed by Eq. (10). We use the mean rank of all gold standard words as the evaluation metric.

### 4.4.2 Experiment Results

Table 2 shows the evaluation results of these models for word analogy inference. From the table, we can observe that:

(1) The SAT model performs best among all models, and the superiority is more significant than that on word similarity computation. This indicates that SAT will enhance the modeling of implicit relations between word embeddings in the semantic space. The reason is that sememes annotated to word senses have encoded these word relations. For example, capital and Cuba are two sememes of the word "Havana", which provide explicit semantic relations between the words "Cuba" and "Havana".

(2) The SAT model does well on both classes of Capital and City, because some words in these classes have low frequencies, while their sememes occur so many times that sememe embeddings can be learned sufficiently. With these sememe embeddings, these low-frequent words can be learned more efficiently by SAT.

(3) It seems that CBOW works better than SAT on Relationship class. Whereas for the mean rank, CBOW gets the worst results, which indicates the performance of CBOW is unstable. On the contrary, although the accuracy of SAT is a bit lower than that of CBOW, SAT seldom gives outrageous prediction. In most wrong cases, SAT predicts the word "grandfather" instead of "grandmother", which is not completely non-sense, because in HowNet the words "grandmother", "grandfather", "grandma" and some other similar words share four common sememes while only one sememe of them are different. These similar sememes make the attention process less discriminative with each other. But for the wrong cases

| Word: 苹果("Apple brand/apple") sense1: *Apple brand* (computer, PatternValue, able, bring, SpeBrand) sense2: *duct* (fruit) | | |
|---|---|---|
| 苹果 素有果中王美称（**Apple** is always famous as the king of fruits） | *Apple brand*: 0.28 | *apple*: 0.72 |
| 苹果 电脑无法正常启动（The **Apple brand** computer can not startup normally） | *Apple brand*: 0.87 | *apple*: 0.13 |
| Word: 扩散("proliferate/metastasize") sense1: *proliferate* (disperse) sense2: *metastasize* (disperse, disease) | | |
| 防止疫情扩散 （Prevent epidemic from **metastasizing**） | *proliferate*: 0.06 | *metastasize*: 0.94 |
| 不扩散 核武器条约（Treaty on the Non-**Proliferation** of Nuclear Weapons） | *proliferate*: 0.68 | *metastasize*: 0.32 |
| Word: 队伍("contingent/troops") sense1: *contingent* (community) sense2: *troops* (army) | | |
| 八支队伍 进入第二阶段团体赛（Eight **contingents** enter the second stage of team competition） | *contingent*: 0.90 | *troops*: 0.10 |
| 公安基层队伍 组织建设（Construct the organization of public security's **troops** in grass-roots unit） | *contingent*: 0.15 | *troops*: 0.85 |

Table 3: Examples of sememes, senses and words in context with attention.

of CBOW, we find that many mistakes are about words with low frequencies, such as "stepdaughter" which occurs merely for 358 times. With the help of sememes, these low-frequency words arise no issue for SAT.

## 4.5 Case study

The above experiments verify the effectiveness of our models for WRL. Here we show some examples of sememes, senses and words for case study.

### 4.5.1 Word Sense Disambiguation

In order to demonstrate the validity of Sememe Attention, we select three attention results in training set, as shown in Table 3. In this table, the first rows of three examples are word-sense-sememe structures of each word. For instance, in the third example, the word has two senses, *contingent* and *troops*; *contingent* has one sememe community, while *troops* has one sememe army. The three examples all indicate that our models can estimate appropriate distributions of senses for a word given a context.

### 4.5.2 Effect of Context Words for Attention

We demonstrate the effect of context words for attention in Table. 4. The word "Havana" consists of four sememes, among which two sememes capital and Cuba describe distinct attributes of the word from different aspects.

Here, we list three different context words "Cuba", "Russia" and "cigar". Given the context word "Cuba", both sememes get high weights, indicating their contributions to the meaning of "Havana" in this context. The context word "Russia" is more relevant to the sememe capital. When the context word is "cigar", the sememe Cuba has more

| Word | 哈瓦那("Havana") | |
|---|---|---|
| Sememe | 国都(`capital`) | 古巴(`Cuba`) |
| 古巴("Cuba") | 0.39 | 0.42 |
| 俄罗斯("Russia") | 0.39 | -0.09 |
| 雪茄("cigar") | 0.00 | 0.36 |

Table 4: Sememe weight for computing attention.

influence, because cigar is a famous specialty of Cuba. From these examples, we can conclude that our Sememe Attention can accurately capture the word meanings in complicated contexts.

## 5 Conclusion and Future Work

In this paper, we propose a novel method to model sememe information for learning better word representations. Specifically, we utilize sememe information to represent various senses of each word, and propose Sememe Attention to automatically select appropriate senses in contexts. We evaluate our models on word similarity and word analogy, and results show the advantages of our Sememe-Encoded WRL models. We also analyze several cases in WSD and WRL, which confirms our models are capable of selecting appropriate word senses with the favor of sememe attention.

We will explore the following research directions in future: (1) The sememe information in HowNet is annotated with hierarchical structure and relations, which have not been considered in our framework. We will explore to utilize these annotations for better WRL. (2) We believe the idea of sememes is universal and could be well-functioned beyond languages. We will explore the effectiveness of sememe information for WRL in other languages.

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
