# Peer review of "Improved Word Representation Learning with Sememes"

_ACL 2017 — decision unknown_

[Official Review · Reviewer 1 · rating 4 · confidence 5]
soundness 5 · originality 5 · clarity 5 · impact 3 · substance 4 · appropriateness 5 · meaningful comparison 3 · presentation format Oral Presentation

This work showed that word representation learning can benefit from sememes
when used in an appropriate attention scheme. Authors hypothesized that sememes
can act as an essential regularizer for WRL and WSI tasks and proposed SE-WL
model which detects word senses and learn representations simultaneously.
Though experimental results indicate that WRL benefits, exact gains for WSI are
unclear since a qualitative case study of a couple of examples has only been
done. Overall, paper is well-written and well-structured.

In the last paragraph of introduction section, authors tried to tell three
contributions of this work. (1) and (2) are more of novelties of the work
rather than contributions. I see the main contribution of the work to be the
results which show that we can learn better word representations (unsure about
WSI) by modeling sememe information than other competitive baselines. (3) is
neither a contribution nor a novelty.

The three strategies tried for SE-WRL modeling makes sense and can be
intuitively ranked in terms of how well they will work. Authors did a good job
explaining that and experimental results supported the intuition but the
reviewer also sees MST as a fourth strategy rather than a baseline inspired by
Chen et al. 2014 (many WSI systems assume one sense per word given a context).
MST many times performed better than SSA and SAC. Unless authors missed to
clarify otherwise, MST seems to be exactly like SAT with a difference that
target word is represented by the most probable sense rather than taking an
attention weighted average over all its senses. MST is still an attention based
scheme where sense with maximum attention weight is chosen though it has not
been clearly mentioned if target word is represented by chosen sense embedding
or some function of it.

Authors did not explain the selection of datasets for training and evaluation
tasks. Reference page to Sogou-T text corpus did not help as reviewer does not
know Chinese language. It was unclear which exact dataset was used as there are
several datasets mentioned on that page. Why two word similarity datasets were
used and how they are different  (like does one has more rare words than
another) since different models performed differently on these datasets. The
choice of these datasets did not allow evaluating against results of other
works which makes the reviewer wonder about next question.

Are proposed SAT model results state of the art for Chinese word similarity? 
E.g. Schnabel et al. (2015) report a score of 0.640 on WordSim-353 data by
using CBOW word embeddings.

Reviewer needs clarification on some model parameters like vocabulary sizes for
words (Does Sogou-T contains 2.7 billion unique words) and word senses (how
many word types from HowNet). Because of the notation used it is not clear if
embeddings for senses and sememes for different words were shared. Reviewer
hopes that is the case but then why 200 dimensional embeddings were used for
only 1889 sememes. It would be better if complexity of model parameters can
also be discussed.

May be due to lack of space but experiment results discussion lack insight into
observations other than SAT performing the best. Also, authors claimed that
words with lower frequency were learned better with sememes without evaluating
on a rare words dataset.

I have read author's response.

[Official Review · Reviewer 2 · rating 3 · confidence 3]
soundness 5 · originality 5 · clarity 4 · impact 3 · substance 3 · appropriateness 5 · meaningful comparison 3 · presentation format Poster

- Strengths:

This paper proposes the use of HowNet to enrich embedings. The idea is
interesting and gives good results.

- Weaknesses:
The paper is interesting, but I am not sure the contibution is important enough
for a long paper. Also, the comparision with other works may not be fair:
authors should compare to other systems that use manually developed resources.

The paper is understandable, but it would help some improvement on the English.

- General Discussion:

[Official Review · Reviewer 3 · rating 4 · confidence 4]
soundness 5 · originality 5 · clarity 2 · impact 3 · substance 4 · appropriateness 5 · meaningful comparison 3 · presentation format Oral Presentation

- Strengths:

1. The proposed models are shown to lead to rather substantial and consistent
improvements over reasonable baselines on two different tasks (word similarity
and word analogy), which not only serves to demonstrate the effectiveness of
the models but also highlights the potential utility of incorporating sememe
information from available knowledge resources for improving word
representation learning.
2. The paper contributes to ongoing efforts in the community to account for
polysemy in word representation learning. It builds nicely on previous work and
proposes some new ideas and improvements that could be of interest to the
community, such as applying an attention scheme to incorporate a form of soft
word sense disambiguation into the learning procedure.

- Weaknesses:

1. Presentation and clarity: important details with respect to the proposed
models are left out or poorly described (more details below). Otherwise, the
paper generally reads fairly well; however, the manuscript would need to be
improved if accepted.
2. The evaluation on the word analogy task seems a bit unfair given that the
semantic relations are explicitly encoded by the sememes, as the authors
themselves point out (more details below).

- General Discussion:

1. The authors stress the importance of accounting for polysemy and learning
sense-specific representations. While polysemy is taken into account by
calculating sense distributions for words in particular contexts in the
learning procedure, the evaluation tasks are entirely context-independent,
which means that, ultimately, there is only one vector per word -- or at least
this is what is evaluated. Instead, word sense disambiguation and sememe
information are used for improving the learning of word representations. This
needs to be clarified in the paper.
2. It is not clear how the sememe embeddings are learned and the description of
the SSA model seems to assume the pre-existence of sememe embeddings. This is
important for understanding the subsequent models. Do the SAC and SAT models
require pre-training of sememe embeddings?
3. It is unclear how the proposed models compare to models that only consider
different senses but not sememes. Perhaps the MST baseline is an example of
such a model? If so, this is not sufficiently described (emphasis is instead
put on soft vs. hard word sense disambiguation). The paper would be stronger
with the inclusion of more baselines based on related work.
4. A reasonable argument is made that the proposed models are particularly
useful for learning representations for low-frequency words (by mapping words
to a smaller set of sememes that are shared by sets of words). Unfortunately,
no empirical evidence is provided to test the hypothesis. It would have been
interesting for the authors to look deeper into this. This aspect also does not
seem to explain the improvements much since, e.g., the word similarity data
sets contain frequent word pairs.
5. Related to the above point, the improvement gains seem more attributable to
the incorporation of sememe information than word sense disambiguation in the
learning procedure. As mentioned earlier, the evaluation involves only the use
of context-independent word representations. Even if the method allows for
learning sememe- and sense-specific representations, they would have to be
aggregated to carry out the evaluation task.
6. The example illustrating HowNet (Figure 1) is not entirely clear, especially
the modifiers of "computer".
7. It says that the models are trained using their best parameters. How exactly
are these determined? It is also unclear how K is set -- is it optimized for
each model or is it randomly chosen for each target word observation? Finally,
what is the motivation for setting K' to 2?